# In silico Approach for Validating and Unveiling New Applications for Prognostic Biomarkers of Endometrial Cancer

**DOI:** 10.3390/cancers13205052

**Published:** 2021-10-09

**Authors:** Eva Coll-de la Rubia, Elena Martinez-Garcia, Gunnar Dittmar, Petr V. Nazarov, Vicente Bebia, Silvia Cabrera, Antonio Gil-Moreno, Eva Colás

**Affiliations:** 1Biomedical Research Group in Gynecology, Vall Hebron Institute of Research, Universitat Autònoma de Barcelona, CIBERONC, 08035 Barcelona, Spain; scabrera@vhebron.net (S.C.); agil@vhebron.net (A.G.-M.); 2Luxembourg Institute of Health, L-1445 Strassen, Luxembourg; elena.martinezgarcia@lih.lu (E.M.-G.); gunnar.dittmar@lih.lu (G.D.); petr.nazarov@lih.lu (P.V.N.); 3Gynaecological Department, Vall Hebron University Hospital, CIBERONC, 08035 Barcelona, Spain; vbebia@vhebron.net

**Keywords:** endometrial cancer, prognostic biomarker, uterine cancer, high-risk, bioinformatics, CPTAC, TCGA

## Abstract

**Simple Summary:**

Endometrial cancer (EC) mortality is directly associated with the presence of poor prognostic factors. Molecular prognostic factors have been identified, but none are used in clinical practice due to lack of validation studies. This study aims to validate a set of 255 prognostic biomarkers previously identified in an extensive literature review and explore new prognostic applications by analyzing them in The Cancer Genome Atlas (TCGA) and Clinical Proteomic Tumor Analysis Consortium (CPTAC) databases. A total of 30 biomarkers were validated and associated to a histological type (*n* = 15), histological grade (*n* = 6), FIGO stage (*n* = 1), molecular classification (*n* = 16), overall survival (*n* = 11), and recurrence-free survival (*n* = 5). Our results encourage further studies of understudied biomarkers such as TPX2, and validates already broadly studied biomarkers such as MSH6, MSH2, or L1CAM, among others. Finally, our results present a significant step to advance the quest for biomarkers to accurately assess the risk of EC patients.

**Abstract:**

Endometrial cancer (EC) mortality is directly associated with the presence of prognostic factors. Current stratification systems are not accurate enough to predict the outcome of patients. Therefore, identifying more accurate prognostic EC biomarkers is crucial. We aimed to validate 255 prognostic biomarkers identified in multiple studies and explore their prognostic application by analyzing them in TCGA and CPTAC datasets. We analyzed the mRNA and proteomic expression data to assess the statistical prognostic performance of the 255 proteins. Significant biomarkers related to overall survival (OS) and recurrence-free survival (RFS) were combined and signatures generated. A total of 30 biomarkers were associated either to one or more of the following prognostic factors: histological type (*n* = 15), histological grade (*n* = 6), FIGO stage (*n* = 1), molecular classification (*n* = 16), or they were associated to OS (*n* = 11), and RFS (*n* = 5). A prognostic signature composed of 11 proteins increased the accuracy to predict OS (AUC = 0.827). The study validates and identifies new potential applications of 30 proteins as prognostic biomarkers and suggests to further study under-studied biomarkers such as TPX2, and confirms already used biomarkers such as MSH6, MSH2, or L1CAM. These results are expected to advance the quest for biomarkers to accurately assess the risk of EC patients.

## 1. Introduction

Endometrial cancer (EC) is the fourth most common cancer in women in developed countries and the sixth in terms of mortality [1]. Unlike other cancers, in the last years EC has been rising in both incidence and associated mortality. By 2040, incidence is expected to increase 23% and mortality will rise by 33% worldwide [2]. Although most women diagnosed with EC are in early-stage disease and have a favorable outcome, the mortality increases dramatically for women with recurrent or advanced disease and for women diagnosed with a clinically aggressive tumor [3].

To manage the diagnosis, treatment, and follow-up of EC patients, multidisciplinary evidence-based guidelines on selected clinically relevant questions have been developed and updated over the years by the European Society for Medical Oncology (ESMO), the European Society of Gynaecological Oncology (ESGO), the European Society for Radiotherapy and Oncology (ESTRO) and the European Society of Pathology (ESP) consortiums [3,4]. These guidelines classify EC in different risk groups based on prognostic factors, including histological subtype (endometrioid or non-endometrioid), tumor grade (low, intermediate or high grade), depth of myometrial invasion, cervical involvement, tumor size, lymphovascular space invasion (LVSI), lymph node status (LNS), tumor spread, and recently, based on molecular classification, which subdivides EC in four molecular groups: POLE ultramutated, microsatellite stability instable (MSI) hypermutated, copy-number low (CN-LOW) (microsatellite stable, MSS), and copy-number high (CN-HIGH) (serous-like) [4,5]. 

These prognostic factors should be determined at the moment of diagnosis since they will guide an optimal surgical treatment, which is the cornerstone treatment for EC; and after surgery, to provide the final diagnosis, stratification of the tumor, and guide the adjuvant treatment [3,4]. While prognostic factors are accurately determined by the histopathological examination of the tumor specimen after surgery, they are often inaccurately determined at early steps of the diagnostic process and need to benefit from multiple approaches. Specifically, histological subtype and tumor grade are assessed through the histopathological examination of an endometrial biopsy, while depth of myometrial invasion, tumor size, cervical involvement, tumor spread, and LNS are determined though imaging techniques. 

The histological diagnosis is determined through the histopathological examination of an endometrial biopsy, preferably obtained by aspiration [4]. It should be reviewed by an expert pathologist to improve the accuracy of histological assessment and the reliability of tumor grading. Although conventional pathological analysis is critical for tumor stratification, it suffers from great interobserver variation [4]. A recent meta-analysis showed discrepancies in 33% of cases between preoperative and postoperative grading. Clinically relevant downgrading was reported in 26% and upgrading in 8% of the patient’s samples [6]. Additionally, interobserver variability also depends on the sampling method. Diagnosis performed by aspiration reached 73% agreement, while hysteroscopic biopsies had a significantly higher agreement (89%) in comparison to dilatation and curettage (70%) [6]. Importantly, the subjective pathological analysis may result in either undertreatment or overtreatment of EC patients, which may lead to comorbidities or even life-threatening risks. 

Multiple imaging techniques are used to determine prognostic factors associated with the stage of the disease. Transvaginal ultrasonography (TVUS), or magnetic resonance imaging (MRI) will assess myometrial invasion, tumor size, and cervical invasion, while computed tomography (CT), MRI or positron emission tomography (PET-CT) will evaluate the lymph node status. However, these techniques are not sensitive enough, and they are not able to determine LVSI, which is a strong predictor of nodal metastasis, recurrence, and cancer-specific death [7,8]. Thus, LVSI is not studied at the time of diagnosis although being an important prognostic factor to predict the outcome of the patients.

Lastly, the emerging molecular classification needs to be determined for the risk assessment of EC patients. To simplify the whole molecular classification, the TransPORTEC group proposed a surrogate classification which is highly effective for most of EC cases and is based on the assessment of POLE sequencing, immunohistochemistry of mismatch repair proteins (MMR-IHQ), and immunohistochemistry of the p53 (IHQ-p53) protein to classify EC patients [9,10,11]. Nevertheless, this approach is not broadly implemented in most hospitals due to methodological limitations and controversies for the evaluation of the test results [12,13,14]. 

Considering this inaccurate scenario for determining the risk classification of EC, an important part of EC research is directed to the hunt for biomarkers, particularly to provide accurate information at the time of diagnosis. In *Coll-de la Rubia E et al., 2020* [15], we reviewed 2507 publications ranging from 1991 to February 2020 and compiled a total number of 255 proteins described as prognostic EC biomarkers. Unfortunately, the vast majority of these biomarkers have not been introduced in clinical practice, probably due to a lack of validation in independent studies, reliability or existing evidence. 

In this publication we aim to validate and identify new prognostic applications for the 255 prognostic biomarkers described in *Coll-de la Rubia E et al., 2020* [15] by performing a statistical analysis using the accessible datasets of TCGA and CPTAC studies [5,16]. Validated proteins were combined to achieve higher accuracy to predict patient’s outcome, and their biological significance was investigated. 

## 2. Materials and Methods

### 2.1. Data Source

Expression data profiles of EC patients were collected from the TCGA database though cBioPortal (https://www.cbioportal.org/, accessed on 27 June 2020). The RNA-Seq expression data of 333 EC patients from the Uterine Corpus endometrial Carcinoma (TCGA, Nature 2013) study (TCGA-RNAseq) was used. CPTAC—Uterine Corpus Endometrial Carcinoma data was obtained from LinkedOmics database (http://www.linkedomics.org/login.php, accessed on 10 June 2020). RNA-Seq (CPTAC-RNAseq) and proteomic (CPTAC-proteome) data corresponding to 95 EC patients was used for the analysis. Table 1 details the clinical information of the EC patients used for the analysis of this study.

### 2.2. Data Processing and Identification of Differentially Expressed Genes (DEGs) and Proteins (DEPs)

The 255 proteins identified in *Coll-de la Rubia E et al., 2020* [15] were subtracted from the three datasets of expression data, which were separately analyzed using the *limma* and *reportROC* packages of R software. The criteria of false discovery rate (FDR) adjusted *p*-value < 0.25, | logFC | > 1, and Area Under the ROC Curve (AUC) > 0.75 were applied to screen the DEGs and DEPs (DEG/Ps). The DEG/Ps that were overlapping in the TCGA-RNAseq and [CPTAC-RNAseq OR CPTAC-proteome] EC datasets were named as validated biomarkers.

### 2.3. Survival Analysis

We used the TCGA dataset to identify the potential genes with an impact on OS and RFS. DEGs with FDR < 0.05 and AUC > 0.6 at time points of 12, 24, 36 or 48 months were subsequently used to construct the Cox proportional hazards regression model to predict OS and RFS. 

### 2.4. Statistical Analysis

Comparisons between histological types (endometrioid EC vs. non-endometrioid EC) and histological grade (low-grade -G1 and G2- EC vs. high-grade -G3- EC) were performed using T Test. Tukey’s honest significance test was used to perform multiple comparisons for FIGO stage (I vs. II vs. III vs. IV) and for molecular classification (POLE vs. MSI vs. CN-LOW vs. CN-HIGH). An AUC value for each comparison was also calculated. The univariate Cox proportional hazards regression analyses were completed using the *survival* package of R software. AUC values were calculated using *survivalROC* package of R for OS and RFS at time points 12, 24, 36 and 48 months. Risk scores for each patient were calculated as follows:$$Risc\;score=\overset{n}{\underset{i=1}{\sum }}exp_{i}*coef_{i}$$
where “*n*” is the number of related prognostic genes, “*exp_i_*” is the expression value of the gene *i*, and “*coef_i_*” is the log hazard ratio (LHR) in univariate Cox regression analysis [17]. Then, the median risk value was used to divide the patients into high and low-risk groups, while the Kaplan–Meier curve was applied to assess the survival difference between the two groups using the log-rank test. Receiver operating characteristic (ROC) curves for assessing the sensitivity and specificity of the prognostic signatures was generated using the *survivalROC* package implemented in R. 

### 2.5. Functional Analysis of DEG/Ps, Interactions, and Tractability Information

We used the Panther database to identify the biological processes and pathways associated with the DEGs. A FDR of < 0.05 was considered as statistically significant [18]. We reported the subcellular location of each protein using UNIPROT [19]. The potential relationship between DEGs encoding proteins was analyzed using the STRING database [20]. Finally, to assess the current available drugs against our DEG/Ps we used the Open Targets Platform [21].

## 3. Results

### 3.1. Study Workflow

We recently used an exhaustive literature revision to compile a total number of 255 proteins as potential prognostic protein biomarkers [15]. These were defined as proteins which have been associated with one or more known clinical prognostic EC factors, including histological subtype, tumor grade, depth of myometrial invasion, cervical involvement, tumor size, LVSI, LNS, tumor spread; as well as the molecular classification, recurrence and/or survival [15]. Most of these biomarkers have not been validated in independent studies, jeopardizing their implementation in the clinical practice. To validate the potential of those proteins as EC prognostic biomarkers and unveil novel potential prognostic associations, we performed an in silico analysis of those proteins in 428 EC patients belonging to the CPTAC and TCGA studies. The workflow of this study is depictured in Figure 1. Briefly, the 255 biomarkers were assessed in the RNA-Seq data of the TCGA and CPTAC datasets, in addition to the proteomic data of the CPTAC dataset. The most relevant prognostic factors, which are histological type, histological grade, FIGO stage, and molecular classification, were analyzed using a differential expression analysis and the calculation of the area under the ROC curve (AUC) values. Additionally, OS and RFS were assessed using a Cox analysis. Statistically significant biomarkers were identified for each parameter and dataset, and those that appeared significant in at least two datasets were considered as validated biomarkers. Among the 255 potential prognostic biomarkers, only 30 biomarkers were validated, and those were further studied using functional analysis and drug tractability studies.

Among the 30 validated biomarkers, we encountered proteins that were significant for one or multiple prognostic parameters (Figure 2). TPX2 is protein associated with a major number of prognostic parameters, including histological type and grade, molecular classification, and OS and RFS. This protein was previously studied in *Jiang T et al., 2018* where it was associated with worse prognosis [22], and recently its prognostic value in EC was further demonstrated [23,24].

### 3.2. Validated Prognostic Biomarkers in EC

Regarding histological subtype, a total of 20, 36 and 18 genes/proteins were differentially expressed between endometrioid (EEC) and non-endometrioid (non-EEC) histologies in the TCGA-RNAseq, CPTAC-RNAseq, and CPTAC-prot datasets, respectively (Figure 3A). However, only 15 were validated in at least two datasets (Figure 3B–D). From those, eight biomarkers were previously described—CCNE1, CDKN2A, ERBB2, ESR1, L1CAM, PAX8, PIGR, VIM—and seven are newly associated to the histological subtype: BUB1, CDC20, CDKN1A, HMGA1, S100A1, TPX2, UCHL1. The most confident biomarker proteins (present in the three datasets) were ERBB2, L1CAM, PIGR and TPX2. Among the non-endometrioid subtypes analyzed in the TCGA dataset, 62 non-EEC cases were included, 10 cases of which were mixed subtype and 52 serous carcinomas. As seen in Appendix A, while some of the biomarkers behaved similarly between the non-endometrioid histologies, others such as CDC20, CDKN1A, ERBB2, HMGA1, L1CAM and PAX8 were expressed in mixed tumors as a mixture between SEC and EEC, similarly to the nature of these tumors.

Regarding histological grade, six proteins in the TCGA-RNAseq cohort, 21 proteins in the CPTAC-RNAseq cohort, and seven proteins in the CPTAC-prot cohort showed different protein abundances between low-grade (grade 1 and grade 2) and high-grade (grade 3) EC. From those, six proteins were validated in two datasets, and were also previously described in other studies: ASRGL1, ATAD2, CDC20, and TPX2 (Figure 3E–H). Thus, highlighting their importance and the need of further validation of those in further prospective clinical studies.

Regarding FIGO stage, three and 17 proteins showed differential abundances in the TCGA and CPTAC cohorts of patients, respectively, 10 of which were previously described in the literature. However, only PGR was validated in two cohorts of patients and its performance was limited to the comparison between stage I and II (Appendix A). 

Finally, this study permitted identifying a significant number of biomarkers that allow separating between different groups of the molecular classification. A total of 16 proteins (ATAD2, CAPG, CCNE1, CDKN2A, ESR1, HMGA1, L1CAM, MSH2, MSH6, PAX8, S100A1, SCGB2A1, TMEFF2, TPX2, TRA2B, UCHL1) were confirmed, i.e., were statistically significant in at least two datasets (Figure 4, Appendix A). Specifically, L1CAM, ATAD2, CAPG, CNNE1, CDKN2A, ESR1, HMGA1, MSH2, MSH6, PAX8, S100A, TPX2, TRA2B, UCHL1, showed capacity to distinguish between CN-LOW vs. CN-HIGH; L1CAM, CDKN2A, HMGA1, MSH6, TMEFF2, UCHL1 between MSI vs. CN-HIGH; and L1CAM and CDKN2A to differentiate between POLE vs. CN-HIGH subgroups. Remarkably, L1CAM seems to be the most informative biomarker to differentiate the CN-HIGH from the other molecular groups. 

### 3.3. Validated Biomarkers Associated to OS and RFS in EC

Among the 255 biomarkers studied, a total of 11 and 5 genes presented significant correlation with survival rates and recurrence, respectively (FDR < 0.05 and AUC > 0.6 at time points of 12, 24, 36, and/or 48 months) in two datasets (Figure 5A,B). In particular, the genes with significant association to OS were ASRGL1, ESR1, FASN, HDGF, MACC1, MCM6, MCM7, MSH2, MSH6, PTK2, and TPX2, while the ones associated with RFS were ATAD2, BUB1, MSH6, TPX2, and TRA2B. Among them, ASRGL1 and ESR1 were characterized as low risk, while the remaining 14 were categorized as high-risk genes. An increased prediction for OS and RFS was achieved by the development of biomarker panels. This was performed by using all the significant genes associated to OS and RFS in a Cox regression analysis. We used the prognostic signature to calculate a risk score (see equation in materials and methods) for each patient, while the median value was used to divide the patients into a high-risk (*n* = 166), and low-risk groups (*n* = 167) (Figure 5A,B). An 11-protein model reached an AUC of 0.827 to predict OS, while a 5-protein model of RFS reached an AUC of 0.712, both at 48 months’ time.

### 3.4. Biological Significance of the Validated Biomarkers

Gene ontology and KEGG enrichment analysis were used to explore the biological functions of the initial 255 proteins, as well as the subgroup of the 30 proteins that were validated in this study. All of them were significantly associated to the following biological processes: cellular processes, biological regulation, response to stimulus, signaling, and metabolic processes. The set of validated genes had an increased association with reproductive processes (Figure 6A). Additionally, we studied the function of those 30 validated biomarkers. While BUB1, CCNE1, CDC20, CDKN1A, CDKN2A, MCM6, MCM7 and TPX2, played a role in cell-cycle, proteins such as ERBB2, ESR1, L1CAM, PGR, PIGR, PTK2 are molecules involved in the activation cascade that enhance the tumor growth. Interestingly, 14 out of the 30 proteins are described as proteins related to the epithelial-mesenchymal transition (EMT), crucial for the malignant progression [25] (Appendix A).

Regarding the pathways analysis, the 255 proteins had a balanced association with multiple pathways, in contrast to the 30 validated biomarkers having a strong relation with the gonadotropin-releasing hormone receptor pathway, p53 pathway, p53 pathway feedback loops 2, interleukin signaling pathway, cell cycle, and integrin signaling pathway, as shown in Figure 6B. Additionally, while downregulated validated biomarkers were associated to interleukin signaling pathway among others, upregulated biomarkers were associated to other pathways such as cell cycle, cadherin signaling pathway, angiogenesis, integrin signaling pathway, VEGF signaling pathway, Parkinson disease, CCLR signaling map, EGF receptor signaling pathway and FAS signaling pathway. The sub-cellular location of the validated biomarkers was mainly in the nuclear and cytoplasm component (Figure 6C). In Figure 6D, the STRING analysis of all validated biomarkers pointed to MCM6, MCM7, CDC20, CCNE1, MSH2, CDKN1A, CDKN2A, ESR1 and ERBB2 at the core of the interaction network. These are fundamental proteins involved in the most altered described pathways triggering EC: ERK, PI3K, WNT, and transcription signaling pathways. Moreover, these are key pathways widely described in cancer. Thus, to further support our findings, we explored each of our 30 validated biomarkers and their prognostic association between other types of cancer in The Human Protein Atlas. All but ESR1, PAX8, PGR, TMEFF2, were associated with prognosis of breast, cervical, colorectal, head and neck, liver, lung, melanoma, ovarian, pancreatic, renal, or urothelial cancers. Interestingly, proteins such as CCNE1, FASN, HDGF, MCM7, PIGR, PTK2, or TRA2B have been described as having a prognostic relation with some other type of gynecological cancer (breast, cervical or ovarian cancer) (see Appendix A). 

Finally, as part of our literature search, we identified chemical probes that have been developed and are currently in different phases of clinical trials targeting some of our validated biomarkers. Specifically, small molecules for different applications have been developed against AURKB, ERBB2, ESR1, PGR, PLK1, and PTK2, as well as therapeutic antibodies against ERBB2 and VIM. In addition, current treatment for EC includes medroxyprogesterone acetate, hydroxyprogesterone caproate, and megestrol acetate targeting PGR, for inoperable patients or for advanced or recurrent tumors [26,27]. Their effectiveness has been reported to increase with the combined use of estrogenic compounds such as tamoxifen targeting ESR1 [28] (Appendix A).

## 4. Discussion

EC is the most common malignancy of the female genital tract. Its early diagnosis is related to good prognosis and overall survival. During the diagnostic process, accurate identification of prognostic factors is crucial for assessing the preoperative risk of recurrence for each patient and guide the surgical treatment. Moreover, the assessment of prognostic factors after surgery is needed for final staging and to guide the adjuvant treatment. To increase the accuracy and objectivity on the diagnostic process, several studies have discovered and described prognostic biomarker candidates, but none of them have reached the clinical practice, mostly because of a lack of independent validation studies. This study focused on the validation of previously described 255 prognostic EC biomarkers. Here we performed an in silico validation of those biomarkers in two of the currently available molecular EC studies, specifically the RNA sequencing data generated by the TCGA and CPTAC and the proteomic data generated by the CPTAC, which compiled data from a total of 428 EC patients. The results obtained here are based on those datasets, which predominately include white and Non-Hispanic or Latino as the main representation for race and ethnicity, respectively. Additionally, in a significant number of cases race and ethnicity clinical data was not reported (60 and 154 cases, respectively). Knowing that EC impacts differently in the overall survival depending on race [1], the discovery and use of EC biomarkers, as well as other cancer biomarkers, could have a different behavior depending on race. Thus, we encourage the scientific community to further investigate and validate their biomarkers on this issue.

Our results revealed 30 biomarkers that show strong evidence for being prognostic EC biomarkers. Remarkably, only ESR1, PGR, ERBB2, L1CAM, MSH2, and MSH6 have been broadly studied as EC biomarkers in the literature [15]. This study adds 30 proteins that may carry important prognostic information in EC and, therefore, should be prioritized for external validation studies. Most of the validated biomarkers discriminate histological subtype and grade, molecular classification, OS and RFS. Regarding FIGO stage, we could only validate PGR as a good prognostic factor, as described by others [29,30]. Among the most outstanding biomarkers, the spindle assembly factor TPX2 merits further attention since its differential expression allowed for the discrimination of all the prognostic factors studied, except for the FIGO stage. TPX2 is a spindle assembly factor required for normal assembly of mitotic spindles, which mediates AURKA localization to spindle microtubules. It has been studied in a broad range of cancers as prognostic marker, including renal, liver, pancreatic and lung cancers [31]. Importantly, TPX2 was also described as prognostic biomarker in gynecological cancer such as breast cancer [32] or ovarian cancer [33]. Regarding EC, it was identified in two bioinformatics studies [22,34], and it has been further investigated in vitro to demonstrate its prognostic ability [23,24,35,36].

Another highlight in this study is the identification of biomarkers to classify EC according to the molecular classification. This classification has acquired an increasing relevance in this last year since it has been incorporated in the most recent clinical guideline of EC to improve risk assessment [4]. Despite the existence of a simplified classification, the Proactive Molecular Risk Classifier for Endometrial Cancer (ProMisE), this molecular classification is not implemented in all centers due to the technical complexity of analyzing the POLE mutations. Our study identified 16 previously described biomarkers with high discrimination potential regarding the molecular groups. We could accurately separate the CN-HIGH group from the others by using the L1CAM biomarker. L1CAM was found frequently expressed in the CN-HIGH group by *Kommoss FK et al., 2018* [37] and they also showed that the L1CAM status was predictive of worse outcome in tumors with no specific molecular profile. More studies are needed to facilitate the classification of EC patients within this molecular classification, particularly to identify the POLE group and the groups classified as multiple classifiers. 

Related to OS and RFS, we developed an 11-biomarker signature, including ASRGL1, ESR1, FASN, HDGF, MACC1, MCM6, MCM7, MSH2, MSH6, PTK2, and TPX2, to predict OS at 48 months with an AUC = 0.827. This model improved the accuracy for the best performing biomarker alone at time points 12, 24, and 36 months. Contrary to this, the model of 5 proteins, including ATAD2, BUB1, MSH6, TPX2, and TRA2B, that predicts RFS only reached AUC = 0.712 at time point 48 months, and its performance was worse than the best performing biomarker to predict RFS alone for time points 12 and 24 months. MSH6 and TPX2 were included in both models; thus, they are highly likely to have an impact on the outcome of the patients. MSH6 is already a well-described protein required to classify patients in the new molecular classification, specifically the MSI group. TPX2 was highlighted in our study as a protein related to multiple prognostic factors. These findings highlight the importance of designing new studies to assess the prognosis of EC patients. Particularly, the need of studies specifically designed to identify biomarkers that can help in the prediction of EC recurrence, as currently it is not clear which are the proteins highly influencing the recurrence of the patients. 

Moreover, the proteins validated in this study have been described to have an important function in the biology of the tumors, having a role in the key pathways triggering cancer such as ERK, PI3K, WNT, as well as transcription signaling pathways. To support our findings, we also explored the prognostic role of these 30 biomarkers in other types of cancer (such as of breast, cervical, colorectal, head and neck, liver, lung, melanoma, ovarian, pancreatic, renal, or urothelial cancers), finding association of those with favorable or unfavorable outcomes of the patients depending on the protein and the cancer type.

In summary, the 30 validated highlighted in this study have shown prognostic power in at least three independent cohorts of patients and thus, they seem promising candidates that merit further validation. Notably, their differential subcellular location should be considered in the design of clinical studies. The cytoplasmic and cellular membrane proteins such as ASRLG1, ATAD2, L1CAM, PGR, TPX2 or UCHL1 should be explored in tissues, whereas secreted proteins such as HDGF, PIGR, SCGB2A1, or TMEFF2 might be easily measured in biofluids, opening the way for a less invasive diagnostic process. In fact, some of the nuclear and cell membrane validated proteins (MSH2, MSH6 and L1CAM) are already used in clinics to diagnose EC patients [3,38], while secreted proteins, such as PIGR, have diagnostic value in uterine aspirates for discriminating between histological subtypes [39].

Considering all the above mentioned, our results confirm that literature revision and further *in silico* validation of the previously described biomarkers in currently available, broadly documented, and considerable in size cohorts of patients such as the TCGA and CPTAC datasets is a valid approach for prioritizing robust biomarkers in future studies. Furthermore, we encourage researchers to validate the 30 biomarkers described in this study in independent studies to achieve the main goal of improving the risk assessment in EC.

## 5. Conclusions

In summary, this study validates and identifies new potential applications of 30 proteins as prognostic biomarkers to discriminate for histological subtype and grade, FIGO stage, molecular classification, overall and recurrence-free survival. A model combination of 11-protein and 5-protein yield a higher accuracy to predict overall and recurrence-free survival, respectively. Our results permit to advance the quest for biomarkers to accurately assess the risk of EC patients.

## Figures and Tables

**Figure 1 cancers-13-05052-f001:**
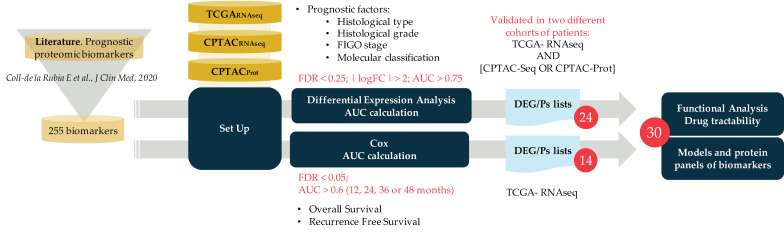
Workflow of the analysis. The 255 biomarkers compiled in *Coll-de la Rubia E et al. 2020* [15] were assessed in two independent cohorts of patients from the TCGA and CPTAC studies. RNA-Seq data from both cohorts and proteomic data from the CPTAC cohort was used. Different prognostic factors were analyzed (histological type, histological grade, FIGO stage, Molecular classification), as well as overall survival and recurrence free survival. Finally, 30 biomarkers were identified as promising for the stratification of EC tumors.

**Figure 2 cancers-13-05052-f002:**
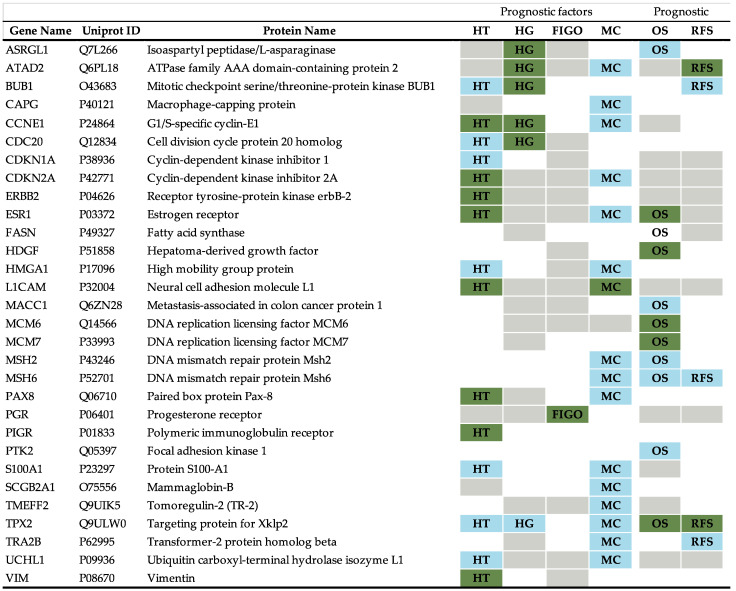
Summary for the 30 validated biomarkers in the analysis. Prognostic factors or prognostic value described in *Coll-de la Rubia E et al., 2020* for each of these biomarkers [15] are highlighted in grey. Additionally, the specific prognostic factor or prognostic value validated in our analysis are indicated for each protein. The color green represents prognostic factors found in both: literature revision and statistical analysis, whereas the color blue represents new prognostic features not described before in literature. HT: histological type; HG: histological grade; FIGO: FIGO stage; MC: molecular classification defined by The Cancer Genome Atlas Network; OS: overall survival; RFS: recurrence-free survival.

**Figure 3 cancers-13-05052-f003:**
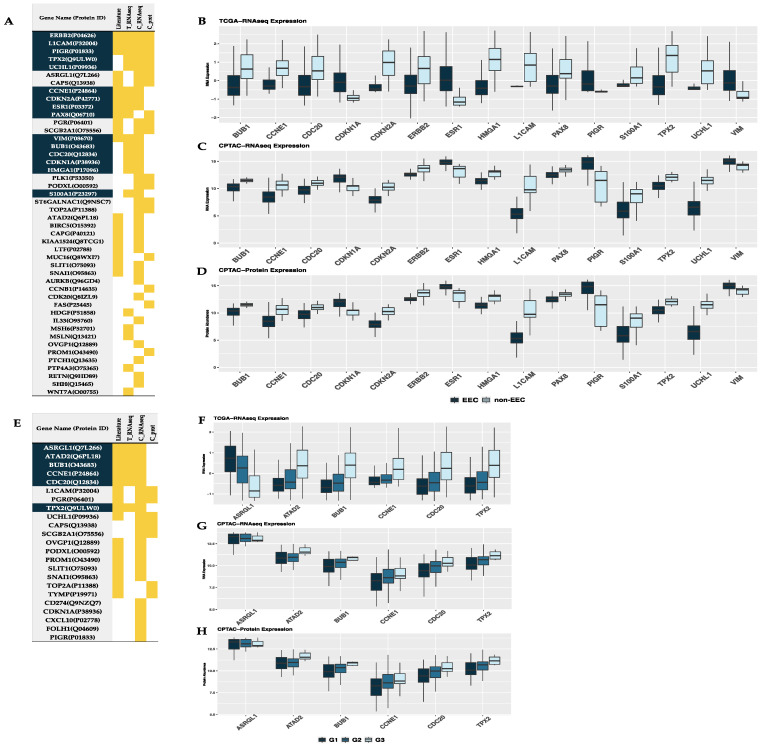
Biomarkers related to histological type and histological grade. (**A**,**E**) Table of the proteins that were found differentially expressed between: (**A**) endometrioid (EEC) and non-endometrioid (non-EEC), and (**E**) G1G2 and G3, respectively, in any of the tested cohorts. Highlighted in yellow, the specific cohort in which that protein was found to be differentially expressed between histologies and/or grades. Proteins highlighted in blue are those validated in more than one cohort, and therefore, the ones that we considered as validated biomarkers. (**B**–**D**) Boxplots showing the expression of the 15 validated biomarkers for histological type in each cohort of patients: TCGA RNA-Seq data, CPTAC RNA-Seq data, and CPTAC proteomic data, respectively. (**F**–**H**) Boxplots showing the expression of the six validated biomarkers for histological grade in each cohort of patients: TCGA RNA-Seq data, CPTAC RNA-Seq data, and CPTAC proteomic data, respectively. Literature: literature revision from Coll-de la Rubia E et al., 2020 [15]; T_RNAseq: RNA-Seq data of the TCGA’s cohort; C_RNAseq: RNA-Seq data of the CPTAC’s cohort; C_prot: proteomic data of the CPTAC’s cohort.

**Figure 4 cancers-13-05052-f004:**
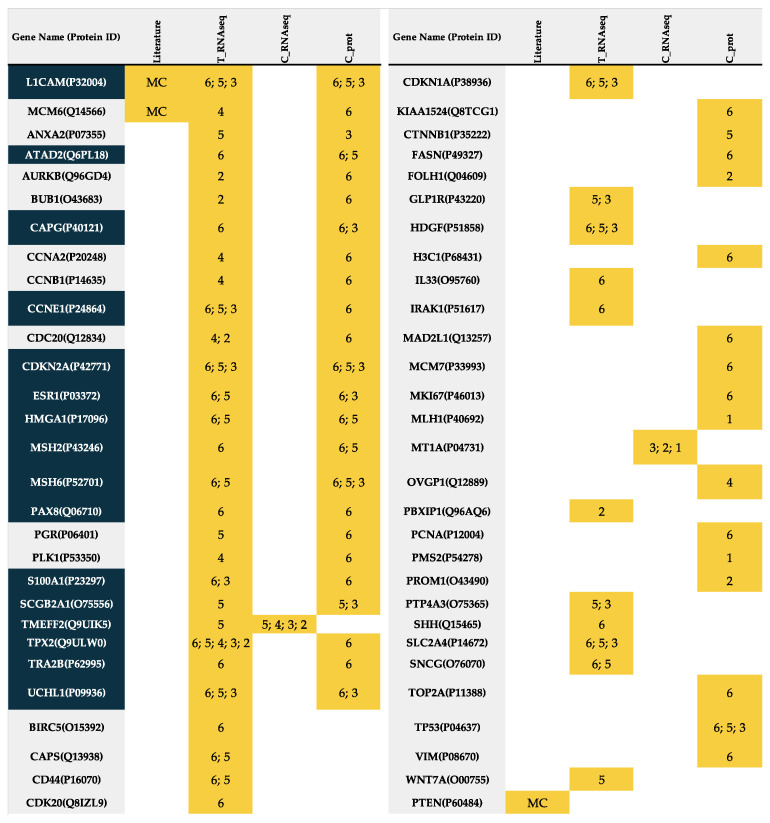
Biomarkers related to the molecular classification. Table of genes/proteins that were found differentially expressed between any of the molecular subgroups in any of the tested cohorts (in grey). Highlighted in yellow, the specific cohort in which each protein was found to be differentially expressed between subgroups and indicated the specific comparison. The numbers indicate the specific comparison where each biomarker was found differential. Proteins highlighted in blue are those validated in more than one cohort. MC: Molecular classification defined by The Cancer Genome Atlas Network; 1: Comparison between POLE mutated vs. microsatellite instability(MSI); 2: Comparison between POLE mutated vs. copy number low (CN-Low); 3: Comparison between POLE mutated vs. copy number high (CN-High); 4: Comparison between microsatellite instability (MSI) vs. copy number low (CN-Low); 5: Comparison between microsatellite instability (MSI) vs. copy number high (CN-High); 6: copy number low (CN-Low) vs. copy number high (CN-High). T_RNAseq: RNA-Seq data of the cohort of the TCGA; C_RNAseq: RNA-Seq data of the cohort of the CPTAC; C_prot: proteomic data of the cohort of the CPTAC.

**Figure 5 cancers-13-05052-f005:**
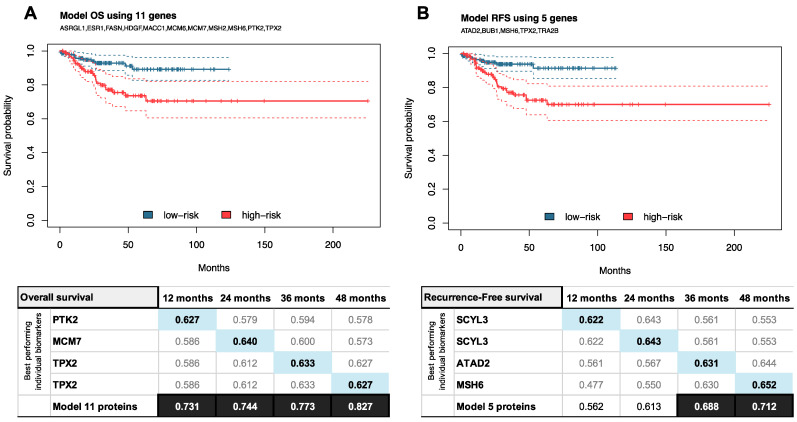
Biomarkers related to overall survival and recurrence free survival and model performance. Best performing individual biomarkers are shown for (**A**) overall survival (OS) and (**B**) recurrence free survival (RFS) for a period of 12, 24, 36, and 48 months, respectively. Additionally, models for both were performed. Regarding prediction of OS, a model of 11 proteins was used, (**A**) while a model of 5 proteins was used to predict RFS (**B**).

**Figure 6 cancers-13-05052-f006:**
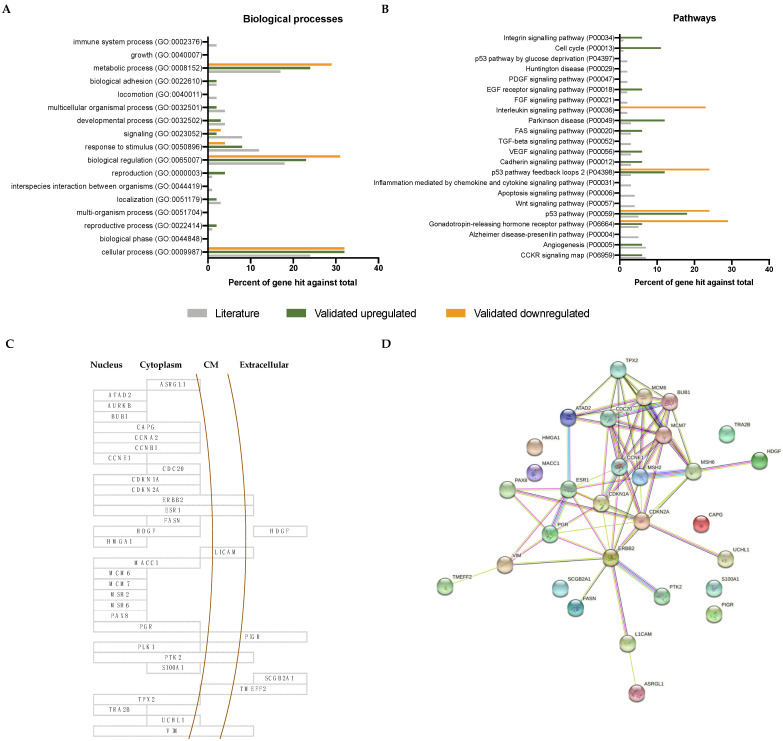
Functional analysis. Top 20 represented biological processes (**A**) and pathways (**B**) from the 255 proteins reviewed in the literature (out of 56 pathways) and all the pathways related to the set of validated proteins. Highlighted in green and yellow are the upregulated and downregulated pathways in relation to the literature, respectively, and in grey, the pathways represented by the proteins compiled in *Coll-de la Rubia E et al., 2020* [15]; (**C**) Subcellular location of the 30 validated prognostic biomarkers; (**D**) String analysis of the 30 biomarkers.

**Table 1 cancers-13-05052-t001:** Clinical, pathological, and molecular information of the patients. Detailed clinical, pathological, and molecular information of the patients included in this study.

Characteristics	TCGA RNA-Seq (*n* = 333)	CPTAC RNA-Seq + Proteome (*n* = 95)
**Age ^1^**		
Mean	63.23 ± 10.91	63.19 ± 9.78
Maximum	90	86
Minimum	33	38
**Histological type**		
Endometrioid	271	83
Serous	52	12
Mixed	10	0
**Grade**		
Grade 1	79	37
Grade 2	90	38
Grade 3	164	8
**FIGO stage ^2^**		
I	226	71
II	19	8
III	70	13
IV	16	3
NA	2	0
**Molecular Classification ^3^**		
POLE	31	7
MSI	92	25
CN-low	110	43
CN-high	78	20
**Overall Survival**		
0: Living	282	36
1: Deceased	51	7

^1^ Age: means and standard deviations are shown. ^2^ FIGO stage: Federation of Gynecologists and Obstetricians for staging. ^3^ POLE: POLE ultramutated; MSI: microsatellite stable instable hypermutated; CN-low: copy number low, endometrioid-like; CN-high: copy number high, serous-like.

## Data Availability

The data presented in this study are available in this article (and Appendix A).

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
