# Peer review of "In silico Approach for Validating and Unveiling New Applications for Prognostic Biomarkers of Endometrial Cancer"

_cancers, 2021, doi:10.3390/cancers13205052_

Round 1

Reviewer 1 Report

This paper represents a nice global approach to the identification of potentially important biomarkers in endometrial cancer prognosis using the TCGA and the CPTAC databases. It's a great springboard for future studies and is a very useful secondary analysis of existing data.

The paper could be strengthened in several ways:

  1. The CPTAC dataset (n=83 endometrioid, n=12 serous cancers) is relatively modest in size, but has depth in the information per subject. Ditto the TCGA (n=333). The description of these cohorts as "large" (eg. line 112) is a bit of an over-statement.
  2. Building on the previous comment, differential expression/protein between endometrioid and non-endometrioid cancer is presented. Yet only 64 women with the serous subtype and n=10 of mixed subtype were included in non-endometrioid. How did this relatively small number of serous cancers as well as the potential misclassification incurred by including mixed subtypes impact the author's ability to identify "significant" differences in expression/protein? This could be added to the discussion.
  3. What is the race/ethnic distribution of these datasets? TCGA is predominately white, but I do not know about CPTAC. Are these results mainly relevant to white women who get endometrial cancer? Could biomarkers be different in  other race/ethnic groups? This is an important discussion point.

Author Response

September 29th, 2021

Dear reviewer,

We appreciate the time and efforts dedicated to review this manuscript. We have addressed all concerns and suggestions raised in here.

Please see the attachment, where you could find a point-by-point response to all your comments.

We believe that the revised version can meet now the journal publication requirements.

We thank you in advance for your kind consideration and look forward to your reply.

Yours faithfully,

Eva Coll & Eva Colas

Co-corresponding authors

Reviewer 2 Report

I would like to thank the editorial board and the authors for the opportunity to review this interesting work.

In the proposed paper the autors validated several prognostic biomarkerkers in endometrial cancer, and this represent a topic of interest.

I would like to thank the editorial board and the authors for the opportunity to review this interesting work. The paper appears rigorous and well written even after careful reading, and in my field of experience I have not been able to find any flaws that prevent the work from being published in its current form. 

Author Response

September 29th, 2021

Dear reviewer,

We appreciate the time and efforts dedicated to review this manuscript and the feedback provided.

We believe that the revised version can meet now the journal publication requirements. We thank you in advance for your kind consideration and look forward to your reply.

Yours faithfully,

Eva Coll & Eva Colas

Co-corresponding authors

Reviewer 3 Report

In this manuscript, Rubia et al, have attempted to validate 255 prognostic biomarkers for endometrial cancer where there is a lack of use of molecular prognostic markers clinically. The authors employed a combination of extensive literature surveys and use of publicly available databases like TCGA and CPTAC to validate the identified biomarkers. From a total of 255 proteins they were able to narrow down potential biomarkers to around 30. Furthermore, they identified TPX2 as a biomarker which warrants further investigation. Their analysis was also able to identify previously used biomarkers like MSH6, MSH2, or L1CAM. This study does contribute to the field and helps to validate some of the prognostic markers that can be applied clinically once more elaborate validation studies have been performed.

There are some concerns with the manuscript which are enlisted below:

  • In the gene ontology and KEGG enrichment analysis, the authors identified biomarkers like MCM6, 7, CDC20 etc. It will be great if the authors could provide brief description about these proteins and/ or their role and if they have been identified as biomarkers for any other cancer type or are being studied in that perspective.
  • In line 308, authors state that they identified chemical probes…..Please clarify if the authors indeed identified/ developed those or just merely found them as part of their literature search
  • Overall, the study lacks any kind of functional validation of the biomarker that they have identified.
  • Typographic errors and misspellings need to be corrected. Provide full-form of some of the abbreviations listed in the Introduction section.

Author Response

(The authors gave the same response as above.)

Reviewer 4 Report

This study revealed the 30 biomarkers of endometrial cancer for distinguishing the subtypes. Figure 3 may be improved to highlight genes which are involved in epithelial-mesenchymal transition. Conclusion may be revised to add the number of biomarkers related to overall survival modeling in figure 5. 

Author Response

(The authors gave the same response as above.)

Round 2

Reviewer 3 Report

The authors have satisfactorily addressed the comments/ suggestions and the manuscript is very much improved.